# Forecasting the prevalence of overweight and obesity in India to 2040

**Shammi Luhar**[1,2]*, **Ian M. Timæus**[1,3], **Rebecca Jones**[4], **Solveig Cunningham**[5], **Shivani A. Patel**[5], **Sanjay Kinra**[1], **Lynda Clarke**[1], **Rein Houben**[6]

1 Department of Population Health, London School of Hygiene & Tropical Medicine, London, England, United Kingdom, 2 Department of Public Health and Primary Care, University of Cambridge, Cambridge, England, United Kingdom, 3 Centre for Actuarial Research, University of Cape Town, Cape Town, South Africa, 4 Laney Graduate School, Emory University, Atlanta, Georgia, United States of America, 5 Hubert Department of Global Health, Emory University, Atlanta, Georgia, United States of America, 6 Department of Infectious Disease Epidemiology, London School of Hygiene & Tropical Medicine, London, England, United Kingdom

* sl989@medschl.cam.ac.uk

**Data Availability Statement:** The National Family Health Survey data used to support the findings of this study have been deposited in the Measure DHS repository (available at:

## Abstract

### Background

In India, the prevalence of overweight and obesity has increased rapidly in recent decades. Given the association between overweight and obesity with many non-communicable diseases, forecasts of the future prevalence of overweight and obesity can help inform policy in a country where around one sixth of the world's population resides.

### Methods

We used a system of multi-state life tables to forecast overweight and obesity prevalence among Indians aged 20–69 years by age, sex and urban/rural residence to 2040. We estimated the incidence and initial prevalence of overweight using nationally representative data from the National Family Health Surveys 3 and 4, and the Study on global AGEing and adult health, waves 0 and 1. We forecasted future mortality, using the Lee-Carter model fitted life tables reported by the Sample Registration System, and adjusted the mortality rates for Body Mass Index using relative risks from the literature.

### Results

The prevalence of overweight will more than double among Indian adults aged 20–69 years between 2010 and 2040, while the prevalence of obesity will triple. Specifically, the prevalence of overweight and obesity will reach 30.5% (27.4%-34.4%) and 9.5% (5.4%-13.3%) among men, and 27.4% (24.5%-30.6%) and 13.9% (10.1%-16.9%) among women, respectively, by 2040. The largest increases in the prevalence of overweight and obesity between 2010 and 2040 is expected to be in older ages, and we found a larger relative increase in overweight and obesity in rural areas compared to urban areas. The largest relative increase in overweight and obesity prevalence was forecast to occur at older age groups.

https://www.dhsprogram.com/data/available-datasets.cfm). The Study on global AGEing and adult health data have been deposited in the WHO Multi-Country Studies Data Archive (available at http://apps.who.int/healthinfo/systems/surveydata/index.php/catalog/sage). Both of the above data sets are available to download for researchers who satisfy the criteria to access confidential data. The publicly available Sample Registration System data have been deposited in the Office of the Registrar General & Census Commissioner repository (available at: http://censusindia.gov.in/vital_statistics/SRS_Based/SRS_Based.html).

**Funding:** This work was supported by the Economic and Social Research Council (https://esrc.ukri.org) [grant number ES/J500021/1] and the funding was received by SL. The funders had no role in study design, data collection and analysis, decision to publish, or preparation of the manuscript.

**Competing interests:** The authors have declared that no competing interests exist.

## Conclusion

The overall prevalence of overweight and obesity is expected to increase considerably in India by 2040, with substantial increases particularly among rural residents and older Indians. Detailed predictions of excess weight are crucial in estimating future non-communicable disease burdens and their economic impact.

## Background

Approximately 39% of the global adult population were classified as overweight (Body Mass Index (BMI) 25.0–29.9 kg/m$^2$) or obese (BMI > 29.9kg/m$^2$) in 2014; a doubling since 1975[1]. Whereas the prevalence of obesity was 6.4% among women and 3.2% among men in 1975, it had risen to 14.9% and 10.8%, respectively by 2014[1]. In developing countries like India, the increasing prevalence of overweight and obesity has coincided with the demographic and epidemiological transitions, in which mortality and fertility have declined, and lifestyle-related diseases have become more common[2–4].

The prevalence of overweight and obesity in India is increasing faster than the world average. For instance, the prevalence of overweight increased from 8.4% to 15.5% among women between 1998 and 2015, and the prevalence of obesity increased from 2.2% to 5.1% over the same period[5–7]. This fast-paced growth has been accompanied by notable increases in the burden of non-communicable diseases (NCDs). Whereas in 1990 the number of life years lost to disability (DALYs) attributable to communicable, maternal, neonatal and nutritional disorders exceeded that attributable to NCDs in virtually all of India's states, currently the opposite is true[3]. Given the extent of the increase in prevalence of overweight and obesity, and its relationships with NCDs[8], reliably predicting its future prevalence has become increasingly important.

Despite this, few studies have attempted to estimate future trends in overweight and obesity in India. One study that reports on global trends estimated that 27.8% of all Indians would be overweight, and 5.0% obese, by 2030[9]. Another study estimated that around 20% of rural Indian adults will be either overweight or obese by 2030[10]. However, these previous studies have merely extrapolated previous trends in prevalence without accounting for a changing population at risk of becoming overweight or obese which declines as the proportion of the population classified as overweight or obese increases.

Simulation models offer a more sophisticated alternative to the extrapolation of secular trends and may produce more accurate forecasts. For example, as an internally logical system, the population at-risk of becoming overweight or obese is regularly updated at each forecasted time interval. Such models therefore allow the incorporation of the impact on future prevalence of past increases in the incidence of overweight or obesity[11]. Additionally, the logical framework enables the estimation of potential impacts of policy decisions, directed at the incidence of overweight and obesity[11, 12], and identification of at-risk subpopulations[11, 13, 14]. This analysis brings together nationally-representative data from a range of publicly available sources in a dynamic simulation model to forecast the future prevalence of overweight and obesity in India to 2040 among adults aged 20–69 years.

## Methods

### Data

1. **National Family and Health Survey (NFHS)**. The nationally-representative NFHS collects health and demographic data among women aged 15–49 years and men aged 15–54 years. NFHS 3 (2005–06) interviewed 124,385 and 74,369 adult women and men respectively, and NFHS 4 (2015–16) contains data on 625,000 adult women and 93,065 adult men[6, 7].

2. **The Study on global AGEing and adult health (SAGE)**. SAGE Waves 0 (2002–04) and 1 (2007–10) contain longitudinal health and demographic data on people aged 50 or more years from six states which are believed to be nationally-representative [15]. Wave 0 collected health information on 2559 adults aged 50 or more years, whereas Wave 1 collected data on approximately 3000 men and 3000 women aged 50 or more years.

3. **Sample Registration System (SRS)**. The SRS reports sex- and residence-specific abridged life tables by five-year age groups for each state for every year between 1997 and 2015[16–21]. The SRS dually records deaths using representative samples from across the country [22].

4. **United Nations World Population Prospects 2019 and World Urbanization Prospects 2018**. The 2019 round of the World Population Prospects includes population projections and estimates by the Population Division of the Department of Economic and Social Affairs of the United Nations Secretariat[23]. The Division uses the cohort-component method for each country and major geographical region to produce population projections under a number of different future fertility scenarios. Separate urban and rural projections that are consistent with the national projections are reported by the UN World Urbanization Prospects 2018[24].

### Model inputs

From these data sources, we extracted the following model inputs for the age, sex and residence-specific forecasts of overweight prevalence in India:

**Age-specific prevalence of overweight and obesity.** We estimated the prevalence of overweight and obesity among individuals aged 20–49 using the BMI variable in NFHS-3 and NFHS-4, whereby individuals with a BMI>24.9kg/m$^2$ and <30.0kg/m$^2$ were classified as overweight, and those with a BMI>29.9kg/m$^2$ were classified as obese following the World Health Organization's (WHO) recommendations[8]. Pregnant women (5.2% of women in NFHS 3 and 4.4% in NFHS 4) were excluded from our sample as their pregnancy could give misleadingly high BMI scores. We used survey weights to calculate age-specific prevalence accounting for the complex survey design and based the baseline (2010) age-specific prevalence on the mid-point prevalence of the two surveys. Among individuals aged 50–69, we used the BMI variable, and the same cut-offs, in SAGE wave 0 and 1 to estimate the overall age-specific prevalence and applied the overall relative risk of overweight and obesity among urban and rural residents to obtain residence-specific prevalence estimates. The prevalence estimates from the data are included in the S2 File.

**Age-specific incidence of overweight and obesity and age-specific rates of urbanization.** We used the changing prevalence of overweight and obesity between 2005 and 2015 among the population aged 20–49 years to estimate the age-specific incidence of overweight among the underweight/normal weight population, and incidence of obesity among overweight individuals in our baseline year of 2010. We used the iterative intracohort interpolation

procedure[25] whereby the observed changes in overweight status to specific cohorts are translated into age-specific rates for the inter-survey interval (a more detailed explanation is presented in S1 File). The age-specific rates were estimated separately by sex, residency (urban and rural) and age (20–24, 25–29, 30–34, 35–39, 40–44, 45–49 years).

Age-specific rates of urbanization were also calculated by the same method, using the age-specific proportions of the population in urban and rural areas in the two NFHS surveys.

For those aged 50–69, we calculated the incidence of overweight and obesity using longitudinal data from SAGE waves 0 and 1 for men and women separately by dividing the number of incident cases by the person-years of exposure. As information on the exact time an incident case occurred was not available, incident cases were assumed to have occurred at the halfway-point between the two waves. We calculated an overall incidence rate among men and women separately, and indirectly standardized the rates using the age-distribution of obesity incidence from a study in the United States[26] that used data from the Behavioral Risk Factor Surveillance System in order to obtain net rates for the following age groups: 50–54, 55–59, 60–64, and 65–69 years (S1 File).

We fitted a spline to smooth the age-specific incidence rates across the lifespan and used age-specific incidence by the five-year age groups in the final analysis.

**Remission.** We incorporated the potential for individuals to transition from overweight and obesity to lower BMI groups by modelling gross, rather than net incidence rates at all ages. Remission refers to reverse transitions, whereby the simulated population is able to transition from a state of 'Obese' to 'Overweight', and 'Overweight' to 'Not Overweight/Obese'. We used rates of remission that allowed our model with gross rates to closely match the measured age-specific prevalence in 2015 from NFHS-4. To estimate remission in older ages, we applied an odds ratio of remission in older ages (50+ years), relative to younger ages. A prospective study in rural India carried out between 2008 and 2017[27] found an elevated odds of remission from higher to lower BMIs of 1.74 and 2.12 among older aged men and women, relative to younger counterparts.

**Current and future age-, sex- and urban/rural residence- specific mortality rates.** We converted conditional mortality probabilities reported by the SRS to age-specific mortality rates from 1997 to 2013 using standard demographic procedures[28] and used these rates to forecast future mortality to 2040 using the Lee-Carter method[29, 30]. In brief, the Lee-Carter method summarizes a series of sets of age-specific mortality rates for successive periods of time by its average age-schedule, age-specific deviations from the average age-schedule, and the trend in the overall level of mortality over time. The forecast is contingent on the extrapolation of this latter parameter (S1 File).

**Relative risk of dying for those overweight or obese compared with those who are not.** We adjusted the forecasted mortality rates to account for differential mortality between overweight and obese individuals and those who are in lower weight categories. Relative risks of dying, based on BMI group, were adopted from the findings reported in a study that examined the association between BMI and mortality in Mumbai[31]. This study reported relative risks of dying for those who are overweight excluding obesity (OW), relative to normal (N) weight, those who are obese (OB) relative to normal weight, and those who are underweight (UW) relative to normal weight. The authors report risk ratios, along with confidence intervals, for men and women aged 35–59 and 60 or more, separately. As the study did not calculate risk ratios for individuals aged 20–34, we assumed that the relative risk of dying at 35–59 also prevailed at these ages. We obtained separate relative risks of dying for those who are overweight and obese relative to new reference categories of 'not overweight' and 'not obese' using a basic algebraic approach (S1 File), and subsequently used these relative risks to calculate BMI specific rates of mortality using the population level mortality rate. Below we present an example of the

calculation of obesity-specific mortality rates.

$$m_{a,t}^{Not\ OB} = \frac{m_{a,t}}{(\delta_{a,t}^{OB} * R_a) - \delta_{a,t}^{OB} + 1}$$

$$m_{a,t}^{OB} = m_{a,t}^{Not\ OB} * R_a$$

Where $\delta_{x,t}^{OB}$ refers to the prevalence of obesity in age group *a* at time *t*, and $R_a$ is the relative risk of dying among obese adults relative to non-obese counterparts aged a.

**Age-, sex- and urban/rural residence-specific population in 2010.** Estimates of the 2010 urban and rural population were taken from the World Urbanization Prospects[24] and disaggregated using the average age-group and sex structure of urban and rural populations separately, which we obtained from the NFHS-3 and NFHS-4[6, 7].

**Population aged 20–24 entering the simulation at every interval.** The new entrants aged 20–24 that join the urban and rural populations in each time interval were estimated using the projected population aged 20–24 from the medium fertility projection scenario by the United Nations' World Population Prospects[23] and split into the projected proportions of the population in urban and rural areas from the World Urbanization Prospects[24].

## The model

We estimated the future prevalence of overweight and obesity through 2040 using an age-stratified simulation model based on a system of multi-state lifetables[32], that moved individuals through mutually-exclusive health states depending on our estimated transition rates as they age. The model operated in discrete time, estimating the prevalence of overweight and obesity separately among men and women in urban and rural areas separately, at five-yearly intervals between 2010 and 2040. The system of multi-state lifetables is shown in Fig 1.

Most epidemiological studies apply transition probabilities to the population at risk of a transition at the beginning of a time period to determine the distribution of the population across health states in a succeeding time period, without taking account of a changing population at risk within a time interval. This is due to individuals being able to enter and re-enter a

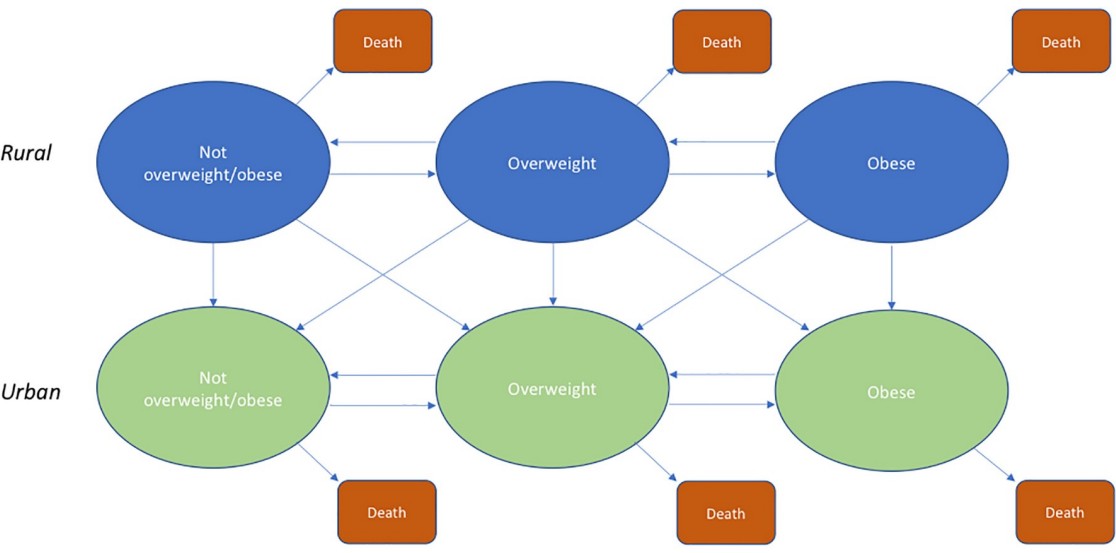

**Fig 1. Compartmental model of forecasted overweight and obesity prevalence in India.**

particular BMI group within a time interval. In order to sufficiently account for this, we employed a multi-state lifetable system first developed by Schoen and Nelson (1974) who addressed questions about flows in and out of marriage in the UK and USA. Rather than work with transition probabilities derived from the rates, this approach to forecasting health states uses the rates directly. A detailed description of this method is included in S1 File.

## Assumptions

Firstly, between 2005 and 2015 the pace of increase in prevalence of overweight and obesity is faster in rural populations than in urban areas. As the prevalence of overweight and obesity among the 20–24 population is not determined within the model, we assumed that the rate of increase in this age group observed from the overweight and obesity prevalence in the NFHS data decreased and converged towards a 0% increase by 2040, so as not to overinflate our estimates. Additionally, our baseline forecasts, assumed that the empirically estimated over-weight and obesity incidence for each demographic group for the baseline year (2010) applied throughout the forecasting period. This assumption provides a clear and easily interpretable counterfactual scenario against which to compare other scenarios whereby incidence is allowed to vary over the forecast period. For simplicity, we assumed that there is no migration in and out of India. Finally, it was assumed that the rate of urbanisation measured between 2005 and 2015 prevailed throughout the forecast period.

## Uncertainty analyses

To obtain uncertainty bounds for our estimates we simultaneously selected random preva-lence, incidence and mortality rates from the distributions that informed their uncertainty. We repeated the simulation 5000 times and we reported the median estimate as the final point estimate, whilst the range of estimates for each population subgroup informed our uncertainty bounds. The analysis was conducted in R version 3.5.1.

## Sensitivity analyses

The future incidence of overweight may continue to increase due to economic development cre-ating an increasingly obesogenic environment. To explore the implications of this potential trend, we included additional scenarios. Scenario 1 involved examining the effect on future prevalence the incidence parameter increasing at a constant annual rate of 1%. In Scenario 2, we examined the effect on future overweight and obesity of the urbanization rate being set at its upper confidence bound throughout the forecast period. Finally, Scenario 3 examined the extent to which the total prevalence of overweight and obesity prevalence would change if no further urbanization were to take place to 2040. Although unrealistic, this provides an understanding as to the extent to which the future increase in prevalence is driven by future urbanization.

We also performed additional analysis using the South Asian BMI cut-offs values. Some advocate the use of these BMI cut-offs due to a stronger positive association between BMI and body fat observed in South Asians compared to White Caucasians, and consequently an ele-vated disease risk at lower BMI levels[33, 34]. Under this assumption, a BMI between 23.0 kg/m$^2$ and 27.5 kg/m$^2$ was used to define individuals who are overweight, and a BMI greater than 27.5 kg/m$^2$ was used to define obesity[35].

## Ethics statement

The analysis of secondary data was approved by the London School of Hygiene & Tropical Medicine's Research Ethics Committee (ref: 16190).

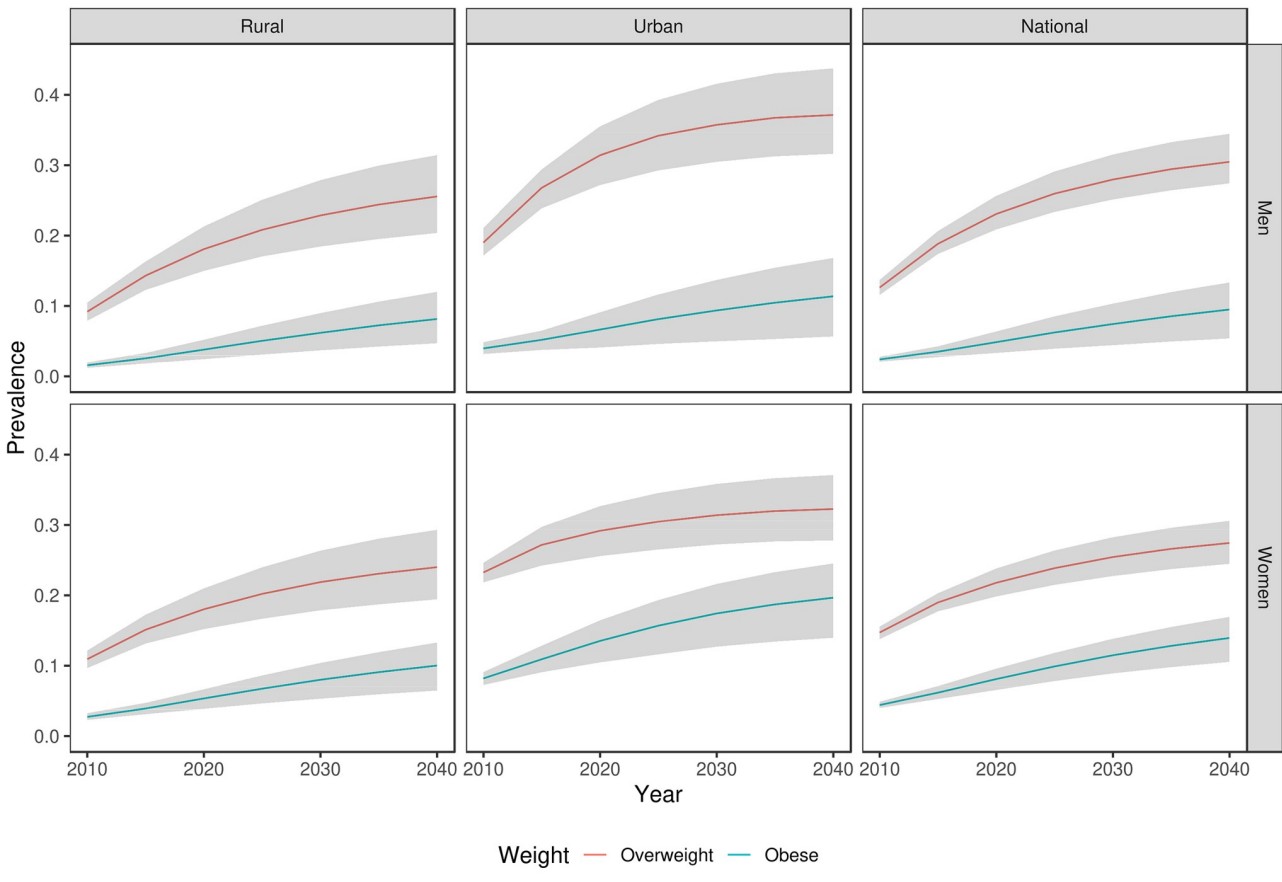

**Fig 2. Forecasted prevalence of overweight and obesity at ages (20–69 years) 2010–2040.**

## Results

Nationally, our model estimates that the prevalence of overweight among women will increase from 14.7% (13.7%-15.5%) to 27.4% (24.5%-30.6%) between 2010 and 2040, whereas the prevalence of obesity is forecasted to increase from 4.4% (4.0%-4.9%) to 14.0% (10.5%-16.9%) over the same period (Fig 2). Among men, the prevalence of overweight and obesity is forecasted to increase from 12.6% (11.6%-13.7%) and 2.4% (2.1%-2.8%) in 2010 to 30.5% (27.4%-34.4%) and 9.5% (5.4%-13.3%), respectively, by 2040 (Fig 2).

The prevalence of overweight and obesity is forecasted to remain higher in urban areas, compared with rural areas, reaching 32.3% (27.8%-37.1%) and 19.7% (14.0%-24.5%), respectively among urban women and 37.1% (31.6%-43.8%) and 11.4% (5.7%-16.8%), respectively, among urban men by 2040. However, the relative increase will be larger in rural areas, where the baseline model forecasts that the prevalence of obesity among women will be 4 times higher in 2040 than in 2010 in rural areas, compared to a 2.2 times higher prevalence of obesity in urban areas over the same period.

The model also predicts larger increases in the prevalence of overweight and obesity in older age groups. Using the broad age groups in Tables 1 and 2, we find that, for example, among men, the prevalence of overweight in urban areas among 55-69-year-olds is predicted to almost quadruple from 10.8% (8.4%-13.6%) to 38.5% (31.3%-48.0%) between 2010 and

**Table 1. Forecast percentage of overweight and obese in the population to 2010–2040 (men).**

| Weight | Residence | Year | 20–34 | | | 35–54 | | | 55–69 | | | All | | |
|---|---|---|---|---|---|---|---|---|---|---|---|---|---|---|
| | | | Point est. | Lower | Upper | Point est. | Lower | Upper | Point est. | Lower | Upper | Point est. | Lower | Upper |
| Overweight | Rural | 2010 | 8.4 | 6.6 | 10.2 | 12.3 | 10.0 | 14.5 | 4.7 | 3.5 | 5.9 | 9.2 | 7.9 | 10.5 |
| | | 2020 | 14.5 | 11.7 | 17.9 | 21.8 | 17.5 | 25.5 | 18.4 | 14.6 | 22.0 | 18.1 | 15.0 | 21.3 |
| | | 2030 | 17.6 | 14.2 | 21.8 | 26.1 | 20.8 | 31.7 | 26.6 | 20.7 | 31.9 | 22.9 | 18.5 | 27.9 |
| | | 2040 | 19.0 | 15.3 | 23.6 | 28.1 | 22.3 | 34.7 | 30.5 | 23.6 | 37.0 | 25.6 | 20.4 | 31.4 |
| Obese | Rural | 2010 | 1.2 | 0.8 | 1.7 | 2.3 | 1.7 | 2.9 | 0.8 | 0.2 | 1.3 | 1.6 | 1.2 | 2.0 |
| | | 2020 | 2.6 | 1.4 | 3.8 | 5.4 | 3.6 | 7.6 | 3.1 | 1.8 | 4.4 | 3.8 | 2.4 | 5.2 |
| | | 2030 | 3.7 | 1.9 | 5.4 | 8.5 | 5.1 | 12.6 | 6.4 | 3.4 | 9.7 | 6.2 | 3.7 | 9.0 |
| | | 2040 | 4.3 | 2.2 | 6.3 | 10.5 | 6.2 | 15.4 | 9.3 | 4.9 | 14.5 | 8.2 | 4.7 | 12.0 |
| Overweight | Urban | 2010 | 15.9 | 14.0 | 17.8 | 25.8 | 22.9 | 29.0 | 10.8 | 8.4 | 13.6 | 19.0 | 17.2 | 21.1 |
| | | 2020 | 24.5 | 21.3 | 27.6 | 38.4 | 33.0 | 44.0 | 30.7 | 25.4 | 36.2 | 31.4 | 27.2 | 35.5 |
| | | 2030 | 28.3 | 24.3 | 32.0 | 41.4 | 35.1 | 49.0 | 37.3 | 30.6 | 45.8 | 35.7 | 30.5 | 41.6 |
| | | 2040 | 30.2 | 25.9 | 34.2 | 42.1 | 35.8 | 50.5 | 38.5 | 31.3 | 48.0 | 37.1 | 31.6 | 43.8 |
| Obese | Urban | 2010 | 3.2 | 2.2 | 4.2 | 5.7 | 4.4 | 7.0 | 1.8 | 0.6 | 2.8 | 4.0 | 3.2 | 4.9 |
| | | 2020 | 5.1 | 3.3 | 7.2 | 8.3 | 4.7 | 12.0 | 6.2 | 2.5 | 9.3 | 6.6 | 4.1 | 9.1 |
| | | 2030 | 6.9 | 4.4 | 9.7 | 11.0 | 5.5 | 16.4 | 10.1 | 3.4 | 16.4 | 9.4 | 5.0 | 13.7 |
| | | 2040 | 7.8 | 5.0 | 11.0 | 13.4 | 6.7 | 19.7 | 12.8 | 3.6 | 22.0 | 11.4 | 5.7 | 16.8 |

2040, whereas the prevalence of overweight in rural areas is predicted to increase from 4.7% (3.5%-5.9%) to 30.5% (23.6–37.0%). On the other hand, our model predicts that younger age groups in our model will experience the smallest absolute increase in the overweight (Figs 3 and 4).

Under the assumption of a 1% annual increase in incidence of overweight and obesity from 2015, we expect the national prevalence of overweight to increase to 29.9% (26.7%-33.7%) by 2040 among women and to 33.1% (29.4%-37.3%) over the same period for men (Fig 5). Over

**Table 2. Forecast percentage of overweight and obese in the population to 2010–2040 (women).**

| Weight | Residence | Year | 20–34 | | | 35–54 | | | 55–69 | | | All | | |
|---|---|---|---|---|---|---|---|---|---|---|---|---|---|---|
| | | | Point est. | Lower | Upper | Point est. | Lower | Upper | Point est. | Lower | Upper | Point est. | Lower | Upper |
| Overweight | Rural | 2010 | 8.0 | 6.6 | 9.5 | 14.9 | 12.5 | 17.0 | 9.3 | 7.8 | 10.8 | 10.9 | 9.7 | 12.1 |
| | | 2020 | 12.6 | 10.4 | 14.8 | 21.4 | 18.1 | 24.9 | 21.9 | 17.8 | 26.3 | 18.0 | 15.2 | 21.0 |
| | | 2030 | 14.7 | 12.1 | 17.4 | 24.5 | 19.9 | 29.4 | 27.6 | 21.3 | 34.5 | 21.9 | 17.9 | 26.3 |
| | | 2040 | 15.6 | 12.9 | 18.4 | 25.9 | 20.9 | 31.4 | 30.0 | 23.3 | 38.1 | 24.0 | 19.4 | 29.3 |
| Obese | Rural | 2010 | 1.7 | 1.2 | 2.2 | 3.9 | 2.8 | 4.9 | 2.5 | 1.6 | 3.5 | 2.7 | 2.3 | 3.3 |
| | | 2020 | 3.3 | 2.3 | 4.2 | 7.1 | 5.3 | 8.8 | 5.7 | 3.0 | 8.3 | 5.3 | 3.9 | 6.6 |
| | | 2030 | 4.4 | 3.1 | 5.7 | 10.0 | 6.9 | 12.7 | 9.7 | 5.1 | 13.9 | 8.0 | 5.3 | 10.4 |
| | | 2040 | 5.0 | 3.5 | 6.4 | 11.9 | 8.0 | 15.4 | 12.6 | 6.3 | 17.7 | 10.0 | 6.5 | 13.3 |
| Overweight | Urban | 2010 | 17.4 | 15.5 | 19.0 | 30.6 | 28.9 | 32.4 | 21.3 | 17.8 | 24.8 | 23.2 | 21.8 | 24.6 |
| | | 2020 | 21.4 | 19.2 | 23.6 | 35.1 | 31.3 | 38.9 | 32.7 | 25.7 | 40.3 | 29.2 | 25.6 | 32.7 |
| | | 2030 | 24.1 | 21.6 | 26.7 | 35.8 | 31.6 | 40.2 | 34.9 | 27.6 | 44.1 | 31.4 | 27.2 | 35.8 |
| | | 2040 | 25.1 | 22.5 | 27.9 | 35.9 | 31.7 | 40.4 | 35.3 | 28.0 | 44.8 | 32.3 | 27.8 | 37.1 |
| Obese | Urban | 2010 | 5.4 | 4.5 | 6.3 | 12.3 | 10.6 | 13.9 | 5.9 | 3.7 | 8.0 | 8.2 | 7.3 | 9.1 |
| | | 2020 | 7.1 | 6.0 | 8.3 | 18.7 | 15.1 | 22.3 | 15.6 | 8.3 | 22.5 | 13.5 | 10.5 | 16.4 |
| | | 2030 | 8.9 | 7.4 | 10.5 | 21.5 | 16.6 | 26.0 | 23.7 | 12.6 | 33.1 | 17.4 | 12.7 | 21.6 |
| | | 2040 | 9.9 | 8.2 | 11.7 | 23.4 | 17.9 | 28.3 | 26.2 | 13.8 | 36.2 | 19.7 | 14.0 | 24.5 |

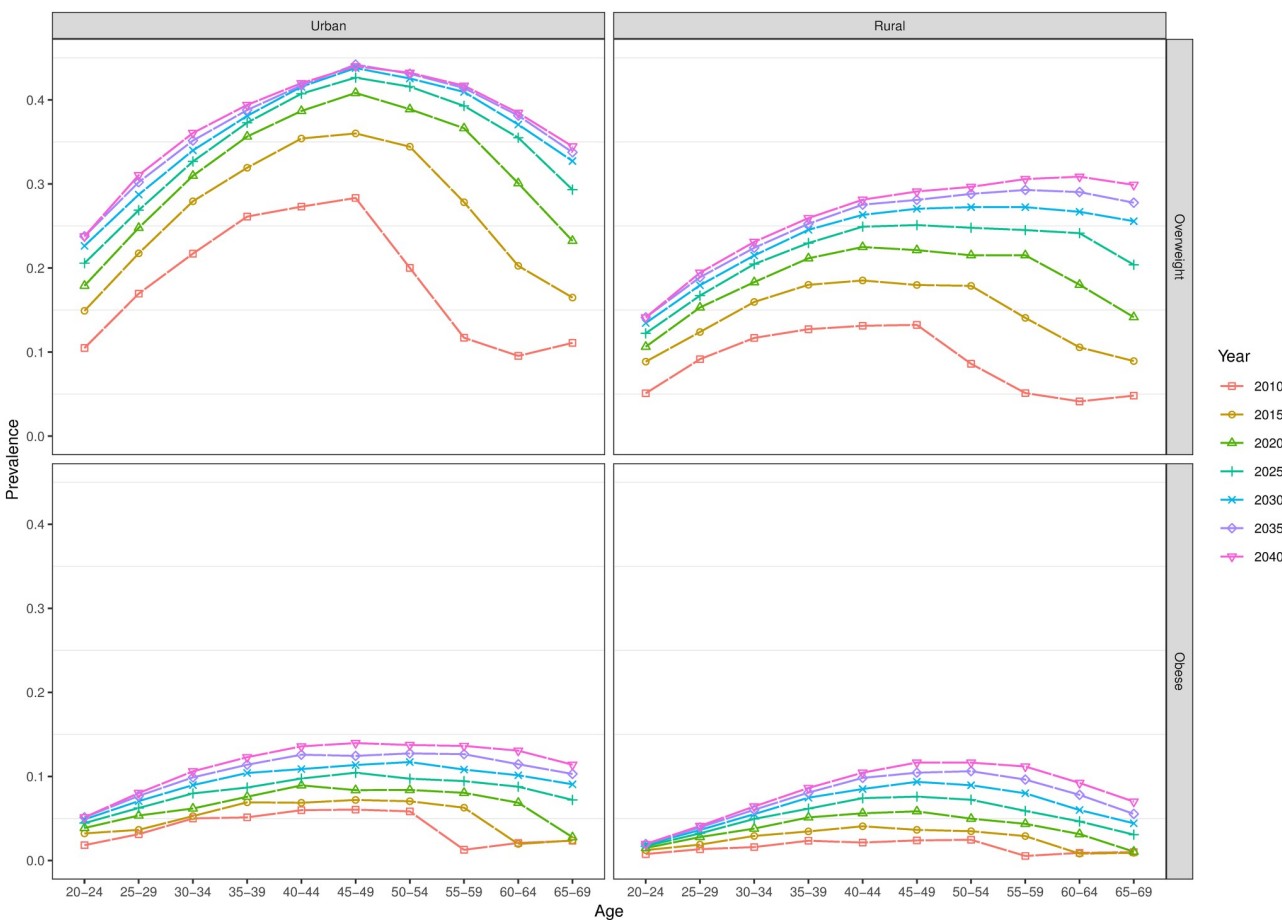

**Fig 3. Forecasted age-specific prevalence of overweight and obesity to 2040 (men).**

the same period, we expect the national prevalence of obesity to increase to 16.9% (11.9%-21.3%) among women and 12.3% (7.8%-17.0%) among men. Under the high urbanization scenario, we find that the future national prevalence of overweight between 2010 and 2040 will increase to 28.4% (25.5%-31.8%) among women, compared to 27.0% (23.7%-30.5%) under an assumption of no further urbanization. The high urbanization scenario for men finds a 1.4% higher percentage point prevalence of overweight among men in 2040, compared to the scenario of no further urbanization. The prevalence of obesity in 2040 does not vary notably between these scenarios (Fig 5).

## Discussion

Overall, we predict that the prevalence of overweight will increase approximately double among Indian adults aged 20–69 years between 2010 and 2040, whilst the prevalence of obesity is expected to increase approximately three-fold over the same period. Specifically, amongst men, we predict that the prevalence of overweight and obesity respectively will reach around 30% and 10%, whilst 27% and 14% of women are expected to be overweight and obese, respectively, by 2040. Our model additionally predicts an ageing distribution of overweight and obesity, with the largest relative increases in prevalence observed among the 55-69-year age group (in this age group the prevalence of obesity among women is predicted to increase almost 6-fold in rural areas and almost 5-fold in urban areas over the forecast period). Whilst

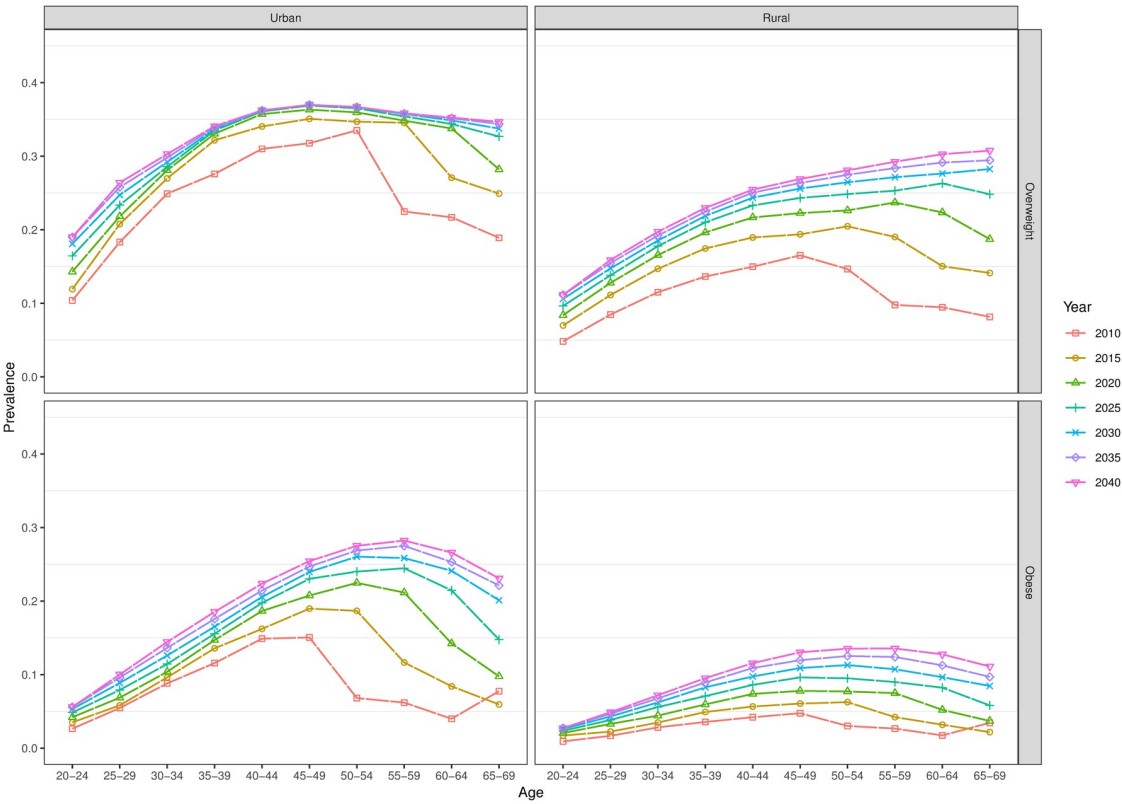

**Fig 4. Forecasted age-specific prevalence of overweight and obesity to 2040 (women).**

prevalence of overweight and obesity is expected to be higher in urban areas throughout the forecast period, we predict larger relative increases in their prevalence in rural areas.

Our forecasting model has a number of limitations. Firstly, we determine the future prevalence of the new cohorts of 20-24-year individuals outside of the model, where we applied a declining rate of increase in prevalence, so as to not grossly inflate future prevalence in this age group to unrealistic levels. Studies have documented increasing overweight prevalence among young adults in India, especially among men and high socioeconomic status individuals[36].

Secondly, we used standard global BMI thresholds over which there is some controversy. Some researchers advocate for using lower BMI thresholds for South Asians[35] due to a higher percentage of body fat among South Asians compared to Caucasians of the same BMI [33, 34]. Some research has documented a nearly 10–15% higher prevalence of overweight among individuals with Asian heritage if Asian-specific cut-offs are used[34]. Others have found no higher risk of mortality among obese Asians compared to obese non-Asians, and advocate for global consistency in the definition of overweight and obesity[9, 37, 38]. We opted to use global cut-offs for this reason and in order to facilitate direct comparison of the predictions with similar forecasting studies in Western countries[39–41]. We sought to remedy this limitation by performing sensitivity analysis using South-Asian BMI cut-offs, and identified potential underestimation of our results (S2 File). For instance, among urban men, we identified a potential underestimation of the 2040 obesity prevalence of around 20 percentage points, suggesting that using global cut-offs may underestimate the future overall public health challenge related to excess weight in India.

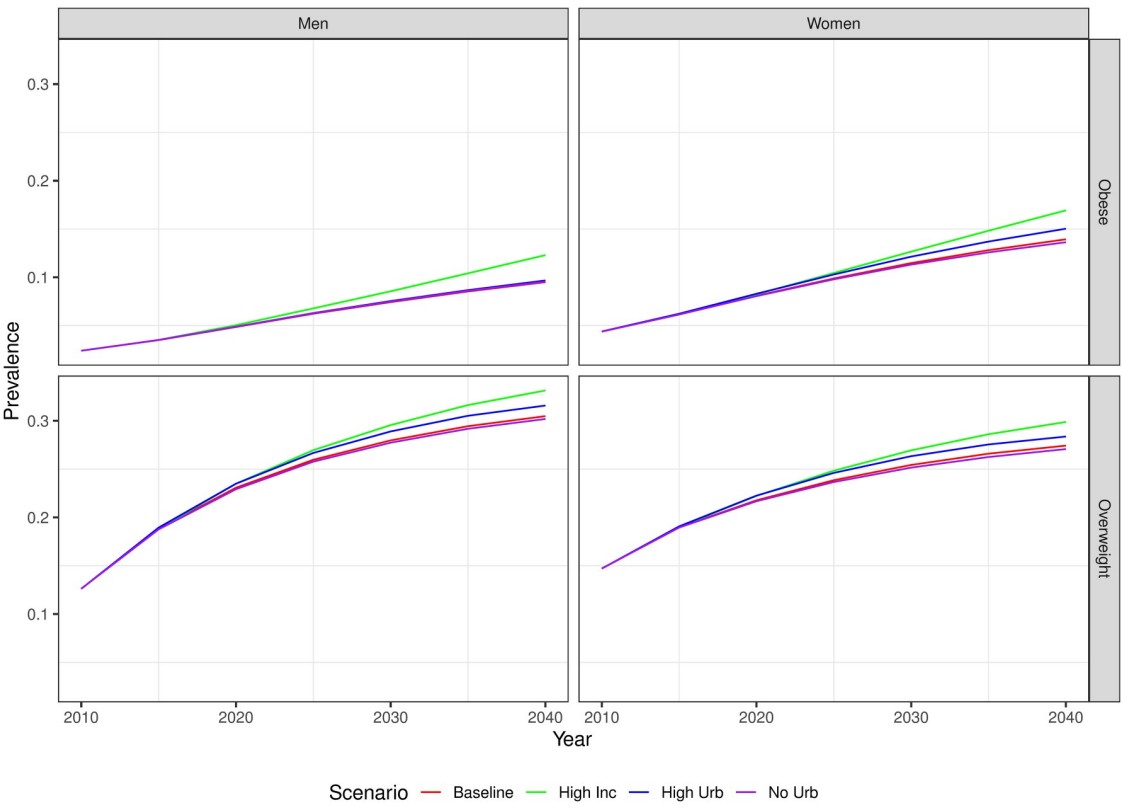

**Fig 5. Forecasted prevalence overweight and obesity to 2040 under the four different scenarios tested.**

Thirdly, our assumption of no migration in and out of India may slightly bias our findings if individuals leaving India to elsewhere are more likely to be overweight or obese than individuals who remain or enter. Any bias attributable to our assumption of zero migration in an out of India is however likely to be negligible as the number of annual net migrants (minus 2.5 million in 2017 according to the World Bank[42]) currently represents less than 1% of the total population[23].

Finally, it would have been desirable to have accommodated the changing socioeconomic patterning of overweight and obesity in India in our forecasts. Studies have shown that rural and lower SEP Indians are at increasing risk of overweight and obesity, with significant variation in this patterning sub-nationally[43,44]. However, due to the uncertainty in how these socioeconomic patterning trends will evolve throughout the forecasting period, the inclusion of socioeconomic status in our model has the potential to, at best, make the predictions marginally more accurate, and at worst, considerably more uncertain, and consequently, we opted for a relatively parsimonious model.

Despite these limitations, our study has a number of strengths. Firstly, we have quantified the future prevalence of overweight and obesity in India using the most recent nationally-representative publicly available data. Our model is able to reflect the changing demographic profile of India in future estimates of the prevalence of overweight and obesity and, in addition, to incorporate future rates of urbanization. Additionally, it models the future age- and sex-specific prevalence of overweight as a function of past and current age- and sex-specific incidence and mortality; reflecting the real-life lag between demographic changes, changes in incidence and mortality and their effect on the overall prevalence at various ages in the future. Unlike

previous studies predicting future prevalence of overweight and obesity in India[9, 10], we forecasted prevalence for age-stratified subgroups as well as generating aggregated forecasts, putting emphasis on demographic groups that are expected to experience particularly high increases in prevalence.

Few studies have attempted to forecast the future prevalence of overweight in India. One study from 2005 predicted that the prevalence of overweight among Indian adults, assuming a continuation of past trends, will increase to 27.8% by 2030, whist the prevalence of obesity is predicted to reach 5.0%[9]. The overweight estimations closely resemble our predictions for men and are slightly above what we predict for women. However, our model predicts a considerably higher prevalence of obesity by 2030, with 11.5% of women and 7.4% of men predicted to be obese by 2030.

Another study, focusing on rural India estimated that the prevalence of overweight will approach 20% among men and just exceed 20% among women aged 18 and over by 2030[10]. We, on the other hand, expect 29.1% of men and 29.9% of women to be either overweight or obese by 2030. The discrepancy between these two separate findings may indeed be due to the different methodologies adopted but is more likely explained the fact that our study included older age groups among whom overweight and obesity prevalence is expected to increase most substantially by 2030.

The differences between our results and previous forecasts of the prevalence of overweight in India may also be explained by our attempts to take into account some of the heterogeneity in the incidence of overweight an obesity and mortality sub-nationally, estimating urban and rural outcomes separately for men and women. Also, instead of making a priori assumptions about the future prevalence of overweight and obesity, for instance a linearly increasing prevalence rate, we model future prevalence as a function of a continuously updated 'population at-risk'. Although we expect our baseline results to be relatively conservative, as we fix age-specific incidence rates over the forecast period, we expect them to be more accurate than previous attempts. This is due to our use of the most up-to-date data, and the fundamental differences in modelling approaches.

The ageing age distribution of overweight prevalence is likely to be driven by a cohort effect. Previous research has reported a peak in the prevalence of overweight in the 40–49 age group in 2005 in India, whereas in more economically developed countries, the prevalence in the same year peaks in the 60–69 age group[9]. Our finding of an older age distribution of both overweight and obesity prevalence in 2040, compared with 2010, may be associated with India's increasing resemblance to higher-income countries in terms of overall prevalence of overweight and economic development. When we tested our forecasts holding future mortality rates at the 2010 level, future prevalence did not notably differ from the forecasts in which future mortality was allowed to decline. Consequently, previous and continuing increases in longevity are not likely to be an important driver of this ageing age distribution of overweight.

We confirmed that our model predictions were very similar to the 2015 age-specific prevalence estimates reported by the NFHS. Another way we assessed the ability of our model to accurately predict future overweight was to compare our output with collected data on overweight prevalence from a data source that was not used in the parameterization of our model. The National Nutrition Monitoring Bureau (NNMB) reports that in 2017 the prevalence of overweight in urban areas was 34% among men. In our model, the prevalence of overweight and obesity combined among urban men is 35%, and the NNMB estimate falls comfortably within our uncertainty bound of 29.5%–40.3%. The NNMB also reports a point estimate of 44.0% prevalence among urban women in 2017, falling within our uncertainty bound of 34.7%– 45.8%, although our point estimate is lower, at 40.4%. We would expect the interval around their estimate to considerably overlap with ours, however this interval was unavailable.

Although NNMB estimates fall comfortably within our uncertainty bound, differences between point estimates can derive from a number of sources. Firstly, different sampling frames are used in the surveys, whereby the NNMB in urban areas selects a sampling frame from under half (16) of Indian states they believe to accurately reflect national trends[45]. Additionally, the NNMB included individuals aged 70 years or more, the majority of whom are likely to be urban women due to their higher life expectancy[21].

We have found that the prevalence of obesity in 2040 is expected to be lower than levels that are currently observed in some of the world's most industrialized economies, implying that India could be susceptible to considerable further increases in obesity prevalence beyond 2040. For instance, a recent survey has found that using the same BMI cut-offs as in our study, 40.4% and 35.0% of women and men in the US, respectively were classified as obese in 2013–14[46], whereas we find that in urban India, a relatively obesogenic environment, 19% of women and 11% of men are likely to be obese by 2040, however, this is one of our most conservative estimates, assuming a constant rate of incidence over the forecast period. Nevertheless, a 1% annual increase in incidence, corresponding to a 35% overall increase in incidence over the forecast period, only leads to a 5 percentage point higher prevalence in combined overweight and obesity by 2040, suggesting that much of the future forecasted prevalence will be determined by the changing demographic profile and background BMI trends of India. Attempts to reduce the forecasted prevalence in 2040 may aim to target a reduction in overweight and obesity incidence, starting among children and adolescents yet to pass through the 20-69-year-old population.

The future task of tackling the increasing disease burden associated with the tripling of obesity prevalence will be particularly challenging in India, given its already high burden of infectious diseases[3], and given that it is soon expected to have the largest population in the world [23]. Obesity is the main risk factor for a range of NCDs, including diabetes. A meta-analysis of prospective cohort studies found a 7.19 times higher risk of diabetes among obese individuals compared to normal weight individuals[47]. Given that people with diabetes are at a high risk of diabetes related complications, including long-term vascular complications affecting the kidneys, heart, and nerves[48], addressing the growing obesity prevalence and ageing pattern of prevalence, is of great urgency. The demand for medical services to tackle the increasing burden of overweight/obesity related diseases is also likely to increase substantially into the near future. Potential interventions include preventative measures such as screening for diabetes among high risk overweight/obese individuals to increase the proportion of people with diabetes that are diagnosed[49]. Further efforts may also wish to improve the provision of already established initiatives, particularly the National Programme for Prevention and Control of Cancer, Diabetes, Cardiovascular Disease and Stroke (NPCDCS), that in-part aims to reduce out of pocket expenditure on diabetes healthcare and promote behavioural and lifestyle improvements that reduce the risk of such diseases[50].

Although the overall prevalence of overweight is expected to be higher in urban areas, our baseline scenario suggests that in urban India future overweight prevalence may begin to plateau during the forecasting period if incidence remains at 2010 levels over the forecast period, while rural areas will continue to experience an increasing prevalence. On the other hand, our model has predicted an almost linear growth in the prevalence of obesity in both urban and rural areas. Irrespective of future incidence or urbanization rate however, our results suggest that a considerably larger proportion of the population in both urban and rural areas will be either overweight or obese by 2040 compared to 2010, driven by the ageing of overweight and obese younger people and increasing prevalence of overweight and obesity in younger ages.

Close monitoring of these populations may be warranted, and interventions to reduce the overall growth in prevalence way wish to target these populations, particularly among

populations susceptible to becoming obese for whom the risk of NCDs is substantially higher [51]. Additionally, health policy planners may wish to pay particular attention to individuals at younger ages to avoid early onset of overweight and the accumulation of overweight prevalence in older age groups.

Given the considerable heterogeneity in customs, diet and economic development between India's states, these forecasts are likely to mask subnational variation. In future work, an examination of how these forecasts may differ at the state level may be particularly useful for health policy planning as the constitution of India devolves the deliverance of health and nutrition policy to the state level[52].

Our model is simple enough to apply to other developing countries with similarly limited data, and its flexibility can be demonstrated by appropriately adjusting the transition rates [53]. Our predictions can also provide the basis of future modelling studies aiming to quantify both monetary costs and future disease burden associated with excess weight in India [54–58].

Our model predicts a considerable increase and an ageing cohort pattern in overweight and obesity across India to 2040, which could have serious implications for future levels of obesity-related diseases, such as diabetes. Initiatives, such as the Integrated National Health Mission [59], which aims to raise overall population health, may wish to use these forecasts to target sub-populations in which the prevalence of excess weight is likely to be highest in the future. Our findings can be extended to quantify the impact of reductions in the incidence of overweight and obesity among certain subgroups and ages. This information may be crucial in estimating the future burden of NCDs, as well as their economic impact.

## Supporting information

**S1 File. Detailed methods—Overweight and obesity forecast model.** This file contains a detailed description of the estimation of model parameters and the procedure followed in the forecasts.
(DOCX)

**S2 File. Supporting information Spreadsheet.** This file contains prevalence of overweight and obesity estimated in the data sets, forecasts by five-year age groups, a table comparing results of forecasting studies in India, and forecasts of overweight and obesity using South Asian BMI cut-offs.
(XLSX)

## Author Contributions

**Conceptualization:** Shammi Luhar, Ian M. Timæus, Sanjay Kinra, Rein Houben.

**Formal analysis:** Shammi Luhar, Ian M. Timæus, Rein Houben.

**Investigation:** Shammi Luhar, Rebecca Jones, Solveig Cunningham, Shivani A. Patel, Sanjay Kinra, Lynda Clarke.

**Methodology:** Shammi Luhar, Ian M. Timæus, Rein Houben.

**Validation:** Shammi Luhar.

**Writing – original draft:** Shammi Luhar.

**Writing – review & editing:** Shammi Luhar, Ian M. Timæus, Rebecca Jones, Solveig Cunningham, Shivani A. Patel, Sanjay Kinra, Lynda Clarke, Rein Houben.

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
