## [Decision Letter · Decision Letter 0]

13 Aug 2019

PONE-D-19-17017

Forecasting the Prevalence of Overweight and Obesity in India to 2040

PLOS ONE

Dear Mr Luhar,

Thank you for submitting your manuscript to PLOS ONE. After careful consideration, we feel that it has merit but does not fully meet PLOS ONE’s publication criteria as it currently stands. Therefore, we invite you to submit a revised version of the manuscript that addresses the points raised during the review process.

We would appreciate receiving your revised manuscript by Sep 27 2019 11:59PM. To enhance the reproducibility of your results, we recommend that if applicable you deposit your laboratory protocols in protocols.io, where a protocol can be assigned its own identifier (DOI) such that it can be cited independently in the future. For instructions see: http://journals.plos.org/plosone/s/submission-guidelines#loc-laboratory-protocols

We look forward to receiving your revised manuscript.

Kind regards,

William Joe

Academic Editor

PLOS ONE

Journal Requirements:

3. Please upload a copy of Figure 6, to which you refer in your text on page 12. If the figure is no longer to be included as part of the submission please remove all reference to it within the text.

Reviewers' comments:

Reviewer's Responses to Questions

**Comments to the Author**

1. Is the manuscript technically sound, and do the data support the conclusions?

Reviewer #1: Yes

Reviewer #2: Yes

2. Has the statistical analysis been performed appropriately and rigorously? 

Reviewer #1: Yes

Reviewer #2: Yes

3. Have the authors made all data underlying the findings in their manuscript fully available?

Reviewer #1: No

Reviewer #2: Yes

4. Is the manuscript presented in an intelligible fashion and written in standard English?

Reviewer #1: Yes

Reviewer #2: Yes

5. Review Comments to the Author

Reviewer #1: In this paper the author has used a system of multi-state life tables to forecast overweight and obesityprevalence among Indians aged 20-69 years by age, sex and urban/rural residence to2040. The author has improved over the existing methodologies employed for similar projection studies by incorporating the change in future risk of becoming overweight by accounting for change in projected population as obesity and overweight increases over time. They report estimates which are close to those provided by NFHS and NNMB, implying that their results are robust. The paper can prove useful for policy makers in the area of health since prevalence of overweight and obesity is associated with numerous non-communicable diseases. However, more information about the values of the inputs used for the study is desirable.

1) The descriptive analyses related with prevalence of obesity for the years considered (From NFHS data and SAGE) for projection exercise can be included before presenting the results. Moreover, results can be presented for 5 year age groups as existing burden is higher for 40+ age groups.

2) A table on comparison of results from projection exercise with that of previous studies for India, their methodology and limitations can be included. The estimated burden can be presented for all the studies with the author’s calculation.

3) A look at the NFHS-4 data reveals that the prevalence of obesity is higher among the upper wealth quintiles. Therefore, projecting the burden separately by wealth quintiles might be a better idea for both rural and urban areas. The risk factor being adjusted for calculating obesity will vary across these groups. The results might change.

4) There is too much variation in BMI levels across States. The percentage of men and women who are overweight is surprisingly low in the most populated States such as Uttar Pradesh and Bihar. Given the dietary habits and pace of urbanization, it might not be desirable to project overall burden with assumptions that urbanization will grow at a stagnant rate. Moreover, use of low BMI thresholds for Asia can be one of the scenarios for the sensitivity analysis.

Reviewer #2: This is a well-written paper (apart from minor typos) and the methods used seem to be reasonable to answer the research question. I have some minor comments:

- Regarding the paragraph on remission it is not clear to me what you mean by this. Please explain what is exactly meant with rates of remission.

- I do not understand the meaningfulness of the comparison of future prevalence rates in India to that of the US (page 19), as obesity rates in the US are already higher also at the baseline. Please explain why you are making this comparison.

-Increases in oweight/obesity do not only result in higher prevalence rates of associated diseases but also in the need for medical care and facilities, which is in the short run more important, as it seems unrealistic with regard to past trends that the trends in overweight and obesity can be reversed in the short term. This could be better elaborated in the discussion.

6. PLOS authors have the option to publish the peer review history of their article (what does this mean?). If published, this will include your full peer review and any attached files.

Reviewer #1: No

Reviewer #2: No

---

## [Author Response · Author response to Decision Letter 0]

2 Sep 2019

Dear Editors,

We thank the reviewers for the comments given on the manuscript titled “Forecasting the Prevalence of Overweight and Obesity in India to 2040” We have edited the manuscript in Track Changes to address their concerns. Below we provide detailed responses to how we have addressed each of the comments. Additionally, the manuscript has been edited to match the journal’s preferred style.

We believe that the manuscript is now suitable for publication in PLoS One.

Best wishes,

Shammi Luhar

Reviewer 1

1) The descriptive analyses related with prevalence of obesity for the years considered (From NFHS data and SAGE) for projection exercise can be included before presenting the results. Moreover, results can be presented for 5-year age groups as existing burden is higher for 40+ age groups.

We agree with the reviewer on this point and have reported the prevalence of overweight and obesity from the data sets in S2 Files. We chose to include this here rather than in the manuscript so as to not clutter the paper. In the section describing the input parameters, we have included the following on page 6 so that readers know they can refer to the Supporting Material for this information: 

“The prevalence estimates from the data are included in the S2 File”.

2) A table on comparison of results from projection exercise with that of previous studies for India, their methodology and limitations can be included. The estimated burden can be presented for all the studies with the author’s calculation.

We agree with the reviewer. We have included a table in the S2 File that compares the results of the other forecasts of overweight and obesity in India, along with their limitations. 

3) A look at the NFHS-4 data reveals that the prevalence of obesity is higher among the upper wealth quintiles. Therefore, projecting the burden separately by wealth quintiles might be a better idea for both rural and urban areas. The risk factor being adjusted for calculating obesity will vary across these groups. The results might change.

Wealth quintiles reported in survey data is a relative measure, and consequently, the definition of what it means to be in each quintile is likely to vary considerably over time. This, therefore, makes its inclusion in the model inappropriate in the author’s opinion. 

The inclusion of a comparative wealth index may be a solution to the issue of comparability over time. However, the association of wealth with overweight and obesity is both complicated, extremely variable by various population subgroups, and has constantly changed over the past two decades. Additionally, the proportion of the population in each quintile of a comparative wealth index is likely to change considerably over time. We refer you to the article below that describes the trends in the association of socioeconomic position with overweight and obesity, and the extent of this variation in these trends sub-nationally[1]. Its inclusion, we believe, would add many extra layers of uncertainty that would detract from the straightforward and relatively elegant model we sought to build. 

1. Luhar, S., Mallinson, P.A.C., Clarke, L. and Kinra, S., 2019. Do trends in the prevalence of overweight by socio-economic position differ between India’s most and least economically developed states?. BMC Public Health, 19(1), p.783.

4) There is too much variation in BMI levels across States. The percentage of men and women who are overweight is surprisingly low in the most populated States such as Uttar Pradesh and Bihar. Given the dietary habits and pace of urbanization, it might not be desirable to project overall burden with assumptions that urbanization will grow at a stagnant rate. 

We take this opportunity to clarify our use of the term urbanization in the context of this study. In this study, we used urbanization to refer to an individual’s propensity to migrate from a rural area to an urban area, and thus do not assume that urbanization will grow at a stagnant rate. 

As urbanization is commonly defined as an increase in the proportion of the total population living in urban areas, assuming that the individual propensity to migrate to urban areas will remain constant over time implies that the growth in the urban population, whilst substantial, will increase at a variable rate. With this assumption, in states with a lower baseline prevalence of overweight/obesity and lower proportion urban, the rate at which the urban population will continue to grow will be faster than in states that have a higher proportion urban at baseline. 

Moreover, use of low BMI thresholds for Asia can be one of the scenarios for the sensitivity analysis.

We agree with the reviewer’s comments that including sensitivity using the BMI cut-off values specific to South Asian populations are desirable. This sensitivity analysis is included in the S2 File, and is also referred to in the text on page 11:

“We also performed additional analysis using the South Asian BMI cut-offs values. Some advocate the use of these BMI cut-offs due to a stronger positive association between BMI and body fat observed in South Asians compared to White Caucasians, and consequently an elevated disease risk at lower BMI levels[2,3]. Under this assumption, a BMI between 23.00 kg/m2 and 27.49 kg/m2 was used to define individuals who are overweight, and a BMI greater than 27.50 kg/m2 was used to define obesity[4].”

We have also included the following on page 17, regarding the limitation of using global-cut-offs:

“We sought to remedy this limitation by performing sensitivity analysis using South-Asian BMI cut-offs, and identified potential underestimation of our results (refer to S2 File). For instance, among urban men, we identified a potential underestimation of the 2040 obesity prevalence of nearly 20 percentage points, suggesting that using global cut-offs may underestimate the future overall public health challenge related to excess weight in India.”

Reviewer 2

1) Regarding the paragraph on remission it is not clear to me what you mean by this. Please explain what is exactly meant with rates of remission.

We agree with the reviewer that this point could benefit from further clarification. In the text we have included the following on page 7:

“Remission refers to reverse transitions, whereby the simulated population is able to transition from a state of ‘Obese’ to ‘Overweight’, and ‘Overweight’ to ‘Not Overweight/Obese.” 

2) I do not understand the meaningfulness of the comparison of future prevalence rates in India to that of the US (page 19), as obesity rates in the US are already higher also at the baseline. Please explain why you are making this comparison.

The way in which I have phrased this is misleading due to my use of the word baseline (which refers to the baseline scenario and not the baseline year). I wanted to convey that even by 2040, India’s prevalence of obesity will not reach the prevalence levels that are currently observed in the United States or other industrialized countries, allowing potential for further increases in future prevalence in India. The sentence has been since altered to the following:

“Our study has found that the prevalence of obesity in 2040 is expected to be lower than levels that are currently observed in some of the world’s most industrialized economies, implying potential for considerable further increases beyond 2040.” 

3) Increases in overweight/obesity do not only result in higher prevalence rates of associated diseases but also in the need for medical care and facilities, which is in the short run more important, as it seems unrealistic with regard to past trends that the trends in overweight and obesity can be reversed in the short term. This could be better elaborated in the discussion.

We agree with the reviewer and have included the following in the discussion to elaborate on this point:

“The demand for medical services to tackle the increasing burden of overweight/obesity related diseases is also likely to increase substantially into the near future. Potential interventions include preventative measures such as screening for diabetes among high risk overweight/obese individuals to increase the proportion of people with diabetes that are diagnosed[5]. Further efforts may also wish to improve the provision of already established initiatives, particularly the National Programme for Prevention and Control of Cancer, Diabetes, Cardiovascular Disease and Stroke (NPCDCS), that in-part aims to reduce out of pocket expenditure on diabetes healthcare and promote behavioural and lifestyle improvements that reduce the risk of such diseases[6].”

1. Luhar, S., Alice, P., Mallinson, C., Clarke, L. & Kinra, S. Do trends in the prevalence of overweight by socio-economic position differ between India ’ s most and least economically developed states ? 1–12 (2019).

2. Misra, A. Ethnic-Specific Criteria for Classification of Body Mass Index: A Perspective for Asian Indians and American Diabetes Association Position Statement. Diabetes Technol. Ther. 17, 667–71 (2015).

3. Stegenga, H., Haines, A., Jones, K. & Professor, J. W. Identification, assessment, and management of overweight and obesity: Summary of updated NICE guidance. BMJ 349, g6608 (2014).

4. Nishida, C. et al. Appropriate body-mass index for Asian populations and its implications for policy and intervention strategies. Lancet 363, 157 (2004).

5. Basu, S. et al. The Health System and Population Health Implications of Large-Scale Diabetes Screening in India: A Microsimulation Model of Alternative Approaches. PLoS Med. 12, 1–21 (2015).

6. Ministry of Health and Family Welfare, G. of I. National Program for Prevention and Control of Cancer, Diabetes, CVD and Stroke( NPCDCS). Directorate General of Health Services Available at: https://dghs.gov.in/content/1363_3_NationalProgrammePreventionControl.aspx. (Accessed: 14th August 2019)

---

## [Decision Letter · Decision Letter 1]

2 Oct 2019

PONE-D-19-17017R1

Forecasting the Prevalence of Overweight and Obesity in India to 2040

PLOS ONE

Dear Mr Luhar,

Thank you for submitting your manuscript to PLOS ONE. After careful consideration, we feel that it has merit but does not fully meet PLOS ONE’s publication criteria as it currently stands. Therefore, we invite you to submit a revised version of the manuscript that addresses the points raised during the review process.

Kindly note that one of the reviewer has suggested revisions to incorporate the vast heterogeneity of India by incorporating features such as urbanization and socioeconomic differentials in the forecasting approach. The authors are requested to incorporate these concerns in the revision or in the authors' reply to the comments.

We would appreciate receiving your revised manuscript by Nov 16 2019 11:59PM. To enhance the reproducibility of your results, we recommend that if applicable you deposit your laboratory protocols in protocols.io, where a protocol can be assigned its own identifier (DOI) such that it can be cited independently in the future. For instructions see: http://journals.plos.org/plosone/s/submission-guidelines#loc-laboratory-protocols

We look forward to receiving your revised manuscript.

Kind regards,

William Joe

Academic Editor

PLOS ONE

Reviewers' comments:

Reviewer's Responses to Questions

**Comments to the Author**

1. If the authors have adequately addressed your comments raised in a previous round of review and you feel that this manuscript is now acceptable for publication, you may indicate that here to bypass the “Comments to the Author” section, enter your conflict of interest statement in the “Confidential to Editor” section, and submit your "Accept" recommendation.

Reviewer #1: (No Response)

Reviewer #2: All comments have been addressed

2. Is the manuscript technically sound, and do the data support the conclusions?

Reviewer #1: No

Reviewer #2: Yes

3. Has the statistical analysis been performed appropriately and rigorously? 

Reviewer #1: No

Reviewer #2: Yes

4. Have the authors made all data underlying the findings in their manuscript fully available?

Reviewer #1: No

Reviewer #2: Yes

5. Is the manuscript presented in an intelligible fashion and written in standard English?

Reviewer #1: Yes

Reviewer #2: Yes

6. Review Comments to the Author

Reviewer #1: I still do not agreee with the assumptions that have been made with respect to urbanization and wealth. Also, the process to arrive the burden is not clear.

Reviewer #2: The points raised in my first review have adequately been addressed. The authors have clarified what they mean with remission. The comparison to the US overweight and obesity rates makes sense now. However the last sentence regarding future potential of obesity beyond 2040 is still a bit misleading. I would suggest not to call it "potential" but e.g. "vulnerability" or "susceptibility". The discussion section has also been elaborated.

7. PLOS authors have the option to publish the peer review history of their article (what does this mean?). If published, this will include your full peer review and any attached files.

Reviewer #1: No

Reviewer #2: No

---

## [Author Response · Author response to Decision Letter 1]

13 Nov 2019

Reviewer #1: I still do not agree with the assumptions that have been made with respect to urbanization and wealth. 

We thank the reviewer for their comment and agree with the reviewer that forecasting the future prevalence of overweight and obesity by separate wealth quintiles would be ideal, however we opted against doing this for the following reasons: 

Firstly, there are inherent inadequacies of a wealth index that is commonly used to capture socioeconomic position in cross sectional surveys in low- and middle-income countries. A wealth index is designed as a relative measure of wealth, rather than an absolute measure. Therefore, the index used to calculate quintiles is specific to the setting and the time period in which it is collected1. Consequently, it cannot be compared over time, and thus cannot be used in the forecasting of overweight and obesity over the specified forecast period. 

Additionally, the changing nature of the association of overweight and obesity with socioeconomic position makes it very difficult to predict in the future which would be set externally to the model. Whilst it is true that survey data shows a higher prevalence of overweight and obesity in higher wealth quintiles, we are unable to firstly determine the proportion of the population that is expected to be in pre-specified wealth quintiles over the forecasting period (2010-2040), and secondly, it is not possible to ascertain what the exact association between overweight/obesity and socioeconomic position will be in the future. A number of systematic reviews2–5, in addition to recent studies in India6–8, have documented and provided evidence that in low and middle income countries, like India, the socioeconomic patterning of overweight and obesity changes considerably over time. The higher prevalence initially observed among high socioeconomic status groups will turn negative along with continued economic development. Due to the likely uncertain nature and considerable subnational heterogeneity in the future socioeconomic patterning of overweight and obesity throughout the forecasting period, we strongly believe that its inclusion in the model would be inappropriate and what results would come out of the model would be uninterpretable due to escalating uncertainty in the data and projections of SES distribution. 

We have also included the following on page 18 under the limitations section: 

“Finally, it would have been desirable to have accommodated the changing socioeconomic patterning of overweight and obesity in India in our forecasts. Studies have shown that rural and lower SEP Indians are at increasing risk of overweight and obesity, with significant variation in this patterning sub-nationally[43],[44]. However, due to the uncertainty in how these socioeconomic patterning trends will evolve throughout the forecasting period, the inclusion of socioeconomic status in our model has the potential to, at best, make the predictions marginally more accurate, and at worst, considerably more uncertain, and consequently, we opted for a relatively parsimonious model.”

References

1. Rutstein SO, Johnson K. DHS Comparative reports No.6. The DHS Wealth Index. Calverton, Maryland, USA; 2004. 

2. Dinsa GD, Goryakin Y, Fumagalli E, Suhrcke M. Obesity and socioeconomic status in developing countries: A systematic review. Obes Rev. 2012;13(11):1067–79. 

3. McLaren L. Socioeconomic status and obesity. Epidemiol Rev. 2007;29(1):29–48. 

4. McLaren L. Socioeconomic status and obesity. Epidemiol Rev. 2007;29(1):29–48. 

5. Monteiro CA, Moura EC, Conde WL, Popkin BM. Public Health Reviews Socioeconomic status and obesity in adult populations of developing countries : a review. Public Health Rev. 2004;82(12):940–6. 

6. Luhar S, Mallinson PAC, Clarke L, Kinra S. Trends in the socioeconomic patterning of overweight/obesity in India: A repeated cross-sectional study using nationally representative data. BMJ Open. 2018;8(10):e023935. 

7. Luhar S, Alice P, Mallinson C, Clarke L, Kinra S. Do trends in the prevalence of overweight by socio-economic position differ between India ’ s most and least economically developed states ? 2019;1–12. 

8. Sengupta A, Angeli F, Syamala TS, Dagnelie PC, Schayck CP van. Overweight and obesity prevalence among Indian women by place of residence and socio-economic status: Contrasting patterns from “underweight states” and “overweight states” of India. Soc Sci Med. 2015;138:161–9. 

In regards urbanisation, this study does not make any assumption that urbanisation (the proportion of the population living in urban areas) will continue at a stagnant rate. In the paper, we use the term ‘urbanisation’ to refer to any one individual’s propensity to migrate to an urban area, not the population level of urbanisation, which is commonly defined as an increase in the proportion of the population residing in urban areas. 

The highly populated and low overweight/obesity states such as Uttar Pradesh and Bihar also have relatively lower proportion of the population in urban areas. In these states, with comparatively lower levels of overweight and obesity, the increase in the proportion of the population residing in urban areas will be relatively fast compared in these low overweight/obesity prevalence areas compared to high overweight/obesity prevalence areas, addressing the reviewer’s initial concerns.

Reviewer #1: Also, the process to arrive the burden is not clear.

We agree with the reviewer that the calculation of the burden was not clear. We have clarified this in the S2 File in a table footnote under the ‘Table of Comparison’ tab. Specifically, we included the following:

“For consistency across the different studies, we calculated the burden using the United Nations World Population Prospects (2017) population projections, and United Nations World Urbanization Prospects (2018), and applying the proportions identified in the studies”

Reviewer #2: The points raised in my first review have adequately been addressed. The authors have clarified what they mean with remission. The comparison to the US overweight and obesity rates makes sense now. However, the last sentence regarding future potential of obesity beyond 2040 is still a bit misleading. I would suggest not to call it "potential" but e.g. "vulnerability" or "susceptibility". The discussion section has also been elaborated.

We thank the reviewer for clarifying that we have adequayely addressed what was meant by remission. We agree with the reviewer’s additional concerns and have amended the sentence in the manuscript.

The original sentence: 

“Our study has found that the prevalence of obesity in 2040 is expected to be lower than levels that are currently observed in some of the world’s most industrialized economies, implying the potential for considerable further increases beyond 2040.”

Has been changed to: 

“We have found that the prevalence of obesity in 2040 is expected to be lower than levels that are currently observed in some of the world’s most industrialized economies, implying that India could be susceptible to considerable further increases in obesity prevalence beyond 2040.”

---

## [Decision Letter · Decision Letter 2]

15 Jan 2020

PONE-D-19-17017R2

Forecasting the Prevalence of Overweight and Obesity in India to 2040

PLOS ONE

Dear Mr Luhar,

Thank you for submitting your manuscript to PLOS ONE. After careful consideration, we feel that it has merit but does not fully meet PLOS ONE’s publication criteria as it currently stands. Therefore, we invite you to submit a revised version of the manuscript that addresses the points raised during the review process.

We would appreciate receiving your revised manuscript by Feb 29 2020 11:59PM. To enhance the reproducibility of your results, we recommend that if applicable you deposit your laboratory protocols in protocols.io, where a protocol can be assigned its own identifier (DOI) such that it can be cited independently in the future. For instructions see: http://journals.plos.org/plosone/s/submission-guidelines#loc-laboratory-protocols

We look forward to receiving your revised manuscript.

Kind regards,

William Joe

Academic Editor

PLOS ONE

Reviewers' comments:

Reviewer's Responses to Questions

**Comments to the Author**

1. If the authors have adequately addressed your comments raised in a previous round of review and you feel that this manuscript is now acceptable for publication, you may indicate that here to bypass the “Comments to the Author” section, enter your conflict of interest statement in the “Confidential to Editor” section, and submit your "Accept" recommendation.

Reviewer #1: All comments have been addressed

2. Is the manuscript technically sound, and do the data support the conclusions?

Reviewer #1: Yes

3. Has the statistical analysis been performed appropriately and rigorously? 

Reviewer #1: Yes

4. Have the authors made all data underlying the findings in their manuscript fully available?

Reviewer #1: Yes

5. Is the manuscript presented in an intelligible fashion and written in standard English?

Reviewer #1: Yes

6. Review Comments to the Author

Reviewer #1: It would have been desirable to include a number of socio-economic determinants such as marital status, job status, income status, smoking, alcohol consumption, sleep duration, psychological factors, dietary intake, and fertility rate which in literature are identified as key determinants of overweight/obesity. But it is also true that it can be very challenging to forecast using these parameters which keep changing. However, the study can be a good starting point for those who want to do further research in this area. At the outset, margin of error should be expected since the values are being projected for such a long period of time. Also, the paper uses the WPP 2017 revision available on the population projection from the UN population projection. Since, the new version (2019) of the WPP and World Urbanization Prospects data is available, I leave it up to the authors to decide whether they want to update the analyses since it can be time consuming and the broad inference might not change.

7. PLOS authors have the option to publish the peer review history of their article (what does this mean?). If published, this will include your full peer review and any attached files.

Reviewer #1: No

---

## [Author Response · Author response to Decision Letter 2]

27 Jan 2020

Reviewer #1: It would have been desirable to include a number of socio-economic determinants such as marital status, job status, income status, smoking, alcohol consumption, sleep duration, psychological factors, dietary intake, and fertility rate which in literature are identified as key determinants of overweight/obesity. But it is also true that it can be very challenging to forecast using these parameters which keep changing. However, the study can be a good starting point for those who want to do further research in this area. At the outset, margin of error should be expected since the values are being projected for such a long period of time. Also, the paper uses the WPP 2017 revision available on the population projection from the UN population projection. Since, the new version (2019) of the WPP and World Urbanization Prospects data is available, I leave it up to the authors to decide whether they want to update the analyses since it can be time consuming and the broad inference might not change.

Response: We thank the reviewer for the feedback on our study and agree that the study can be a good starting point for those who want to do further research in this area. We also thank the reviewer for the opportunity to update our analyses with newly published data. We have since re-run the models with the new data and the changes have been included in the manuscript, graphs, tables and the supplementary information.

---

## [Decision Letter · Decision Letter 3]

7 Feb 2020

Forecasting the Prevalence of Overweight and Obesity in India to 2040

PONE-D-19-17017R3

Dear Dr. Luhar,

We are pleased to inform you that your manuscript has been judged scientifically suitable for publication and will be formally accepted for publication once it complies with all outstanding technical requirements.

With kind regards,

William Joe

Academic Editor

PLOS ONE

Additional Editor Comments (optional):

Reviewers' comments:

Reviewer's Responses to Questions

**Comments to the Author**

1. If the authors have adequately addressed your comments raised in a previous round of review and you feel that this manuscript is now acceptable for publication, you may indicate that here to bypass the “Comments to the Author” section, enter your conflict of interest statement in the “Confidential to Editor” section, and submit your "Accept" recommendation.

Reviewer #1: All comments have been addressed

2. Is the manuscript technically sound, and do the data support the conclusions?

Reviewer #1: (No Response)

3. Has the statistical analysis been performed appropriately and rigorously? 

Reviewer #1: (No Response)

4. Have the authors made all data underlying the findings in their manuscript fully available?

Reviewer #1: (No Response)

5. Is the manuscript presented in an intelligible fashion and written in standard English?

Reviewer #1: (No Response)

6. Review Comments to the Author

Reviewer #1: (No Response)

7. PLOS authors have the option to publish the peer review history of their article (what does this mean?). If published, this will include your full peer review and any attached files.

Reviewer #1: No

---

## [Editor Report · Acceptance letter]

11 Feb 2020

PONE-D-19-17017R3 

Forecasting the Prevalence of Overweight and Obesity in India to 2040 

Dear Dr. Luhar:

I am pleased to inform you that your manuscript has been deemed suitable for publication in PLOS ONE. Congratulations! Your manuscript is now with our production department. 

With kind regards,

on behalf of

Dr. William Joe 

Academic Editor

PLOS ONE